# A systematic review exploring the factors that contribute to increased primary care physician turnover in socio-economically deprived areas

**Jasmine Lee** [1]*, **Evangelos Kontopantelis** [2]

**1** Faculty of Biology, School of Medicine, Medicine and Health, University of Manchester, Manchester, United Kingdom, **2** Faculty of Biology, Division of Informatics, Imaging and Data Sciences, School of Health Sciences, Medicine and Health, University of Manchester, Manchester, United Kingdom

* jasmine.lee@student.manchester.ac.uk

**Data Availability Statement:** All relevant data are within the manuscript and its Supporting Information files.

## Abstract

### Background

The declining trend in the number of primary care physicians worldwide has led to shortages especially within socioeconomically deprived areas. Socioeconomically deprived areas in the context of this review are defined by regions where there are lower levels of income and access to essential services such as primary healthcare compared to other areas. This shortage contributes to a higher incidence of preventable hospital admissions, unnecessarily straining healthcare infrastructure and negatively affecting patient outcomes. Previous studies have often been limited in scope, focusing on isolated factors or specific regions. Therefore, the objective of this systematic review is to synthesise current research to provide a better understanding of the underlying causes of this high turnover, ultimately informing strategies to address the global shortage of primary care physicians.

### Methods

This systematic review followed the guidelines of the Preferred Reporting Items for Systematic Reviews and Meta-Analyses (PRISMA). Refer to S1 Table for the PRISMA 2020 checklist. A comprehensive search was conducted across PubMed (1970 to September 2024) and Embase (1974 to September 2024). The eligibility criteria included quantitative empirical studies that included a measurement of at least one of the factors behind increased primary care physician turnover or retention within socio-economically deprived or disadvantaged areas. However, the included studies were required to employ a specific methodology for classifying or defining socioeconomic deprivation. The references were screened, the studies selected, the data extracted, and the risk of bias assessed using the ROBINS-I tool, with both reviewers in agreement.

### Results

Thirteen studies were identified. The factors measured in the studies driving increased turnover in deprived areas included region of work (n = 7), income (n = 2), burnout (n = 2) and

**Funding:** The author(s) received no specific funding for this work.

social values (n = 2). Some studies found additional challenges specific to socioeconomically deprived areas, such as familial concerns about regional safety, limited employment opportunities for spouses, or personal career development challenges. However, some studies identified increased hours and sickness presenteeism as stronger contributors to burnout. However, this link can be presumed to be stronger in deprived areas due to staffing shortages, though none of the studies in this systematic review have directly measured this correlation. Though longer-term methods of retention within socioeconomically deprived areas included more collaborative working environments and flexible working hours, this can also be applied to benefit healthcare settings across all regions.

## Conclusions

The studies reviewed have consistently highlighted the repeating cycle of persistent staff shortages contributing to an increased turnover rate within disadvantaged areas internationally. Therefore, implementation of targeted policies by governments and healthcare organisations is required to retain primary care physicians within these areas to ultimately improve and standardise patient care.

## Introduction

Primary care physicians, otherwise referred to as general practitioners (GP), or family practitioners in the USA, typically serve as the initial point of contact for individuals seeking medical guidance in primary healthcare institutions [1]. Socioeconomic deprivation was broadly defined based on the parameters of the Index of Multiple Deprivation Score which takes into consideration employment, education and training, health, housing, and living environment deprivation. However, recent studies have shown a concerning downward trend in the number of primary care physicians worldwide, reflected in NHS England with a 7000 primary care physician shortage in 2024 [2]. Research indicates that this shortage is not uniformly distributed throughout England, as regions characterised by a higher level of deprivation tend to experience a higher level of primary care physician scarcity [3]. This aligns with the Inverse Care law, which suggests the population who are most in need of healthcare services are often the least likely to receive them as supply is unable to keep up with need [4]. This trend has continued from its initial establishment of this issue in 1971 till today and is observed internationally, not only in low or middle income countries, but also in countries classified as high income such as the UK and the USA which the included studies within this systematic review focuses on [5]. Therefore, there needs to have a significant boost in GP funding, coupled with a more equitable distribution of resources that prioritises areas with higher levels of deprivation.

Primary care physician turnover refers to the concrete act of employees leaving the practice. Turnover intention refers to the thought of leaving, though it frequently serves as a reliable gauge for actual turnover [6]. A high turnover and a low retention rate exacerbate this downward trend as primary care physicians either transition to another practice, exit the profession, opt for early retirement, or pursue positions with reduced clinical responsibilities [7]. This interrupts the continuity, and thus quality of care for patients. A high primary care physician turnover rate was associated with 1.8 more emergency department attendances for every 100 patients, associated with the 5.2% fewer people being able to see their preferred doctor [8].

Furthermore, Dale et al. [9] found 82% of primary care physicians in England intended to: exit general practice, retire, take a career hiatus, or reduce their clinical hours within the upcoming five years. The study attributes this to the increasing intensity and volume of workload and the increasing expectations placed on general practice by the NHS, government and patients, leading to burnout. This fuels low morale and diminished job satisfaction, hence leading to increased expression of turnover.

Despite the implementation of financial incentives, there is persistently high turnover within socio-economically deprived areas, with one of the main issues being a lack of staff and support [10]. This trend is seen internationally within Canada, Australia and the UK [10–12], with slightly differing methods being offered, ranging from an increase in salary in Canada and Australia, compared to a pay for performance method in the UK. For instance, in Canada, despite the average incomes of physicians increasing from three and a half times to four and a half times the average Canadian salary, a disproportionate migration pattern favouring urban areas with larger populations persist [10]. This is mirrored in Australia, as a hypothetical 10% increase in salary for primary care physicians to work in low socioeconomic status neighbourhoods would most likely result in only 0.8% of primary care physicians relocating [11]. Similarly, the introduction of the Targeted Enhanced Recruitment Scheme in the UK offers primary care physician trainees a £20,000 payment to train in areas deprived of doctors in Scotland, which are often also socio-economically deprived [13]. However, only 21% of doctors reported being influenced by the program [12]. Therefore, this systematic review highlights the need for holistic retention strategies including the recruitment of additional staff to ensure manageable workloads [14]. This also helps increase collaboration between primary care physicians and allied health professionals, reducing professional isolation and increasing retention [15].

Current literature on primary care physician turnover examines individual-level factors which include burnout, income and social values of primary care physicians. They also cover systemic issues such as workforce distribution amongst differing socio-economic areas and the lack of professional job opportunities, which may deter current medical students pursuing the primary care physician speciality, further perpetuating the shortage [16,17]. These studies generally focus on a specific country or area and lack a global picture.

Therefore, this study aims to synthesise existing research on the factors contributing to primary care physician turnover and retention within socio-economically deprived areas to offer a coherent, global perspective.

This in turn allows the examination of how these specific gaps can be amended by policy interventions, although we did not explicitly search for these.

## Methods

### Search strategy

The two electronic databases were last accessed and searched on 15/9/2024 (Table 1). The articles where the full texts were unable to be obtained were excluded. Keyword terms were included within a multi-field search and were based on commonly used terms within the

**Table 1. Table detailing electronic databases searched, and number of papers identified.**

| Database | Papers Identified |
| --- | --- |
| 1ST SEARCH: | |
| PUBMED (1970–2024) | 2562 |
| Embase (1974–2024) | 144 |

realms of primary care physicians, turnover, retention and deprivation. Medical Subject Headings (MeSH) terms were not included in the search.

The following search was performed:

General Pract* OR GP* Primary Care* OR Family Doctor* OR Family Physician* OR Family Pract* OR General Physician* OR General Medical Pract* OR Primary Healthcare* OR Community Doctor* OR Community Health Physician* OR Outpatient Care provider* OR Primary Health Clinician* OR PCP or FMD or FP

AND

turnover OR attrition OR resignation OR workforce movement OR retention

AND

socioeconomic deprivation OR socio-economic deprivation OR socioeconomic inequality OR socio-economic inequality OR socioeconomic disadvantage OR socio-economic disadvantage OR socioeconomic status OR social deprivation OR economic deprivation OR social inequality OR economic inequality OR socioeconomic hardship OR socio-economic hardship OR poverty OR low*income OR disadvantaged population* OR marginali*ed OR social exclusion OR underprivileged OR deprived area* OR inequity OR inequality.

Refer to S1 File for the extraction of the search strategy.

Justification of the search:

The symbol '*' was employed to capture variations of key search terms. For example, the term primary care* included multiple forms such as primary care physician(s), primary care provider(s) and primary care clinician(s). Similarly, general pract* encompassed variations such as general practice(s) and general practitioner(s). This ensured a comprehensive coverage of relevant terms and enabled consistency across the search by capturing different forms of the word.

Additionally, commonly used abbreviations such as GP (general practitioner), PCP (primary care physician), FMD (family medicine doctor) and FP (family physician) were also included to ensure all potential references were covered.

Synonyms such as attrition, resignation and job satisfaction were incorporated to capture a wide range of relevant concepts. Similarly, a broad selection of synonyms for socioeconomic deprivation including socioeconomic disadvantage, poverty, and low income were used to ensure comprehensive coverage of the concept socioeconomic deprivation.

## Study selection

A flow chart demonstrating the selection process is presented in Fig 1.

Figure shows the methodical step by step inclusion and exclusion of articles for this systematic review.

After the removal of the duplicate articles, titles and abstracts were screened and filtered. The first step of screening the title and abstracts were performed twice by one of the reviewers to ensure all the articles removed, were on irrelevant topics or systematic reviews. Then, a filtering of the full text was performed where both reviewers collaborated and were in agreement that the selected articles included in the study were relevant to the topic of the systematic review. Refer to S2 Table for the complete list of articles and the reasons for their exclusion.

## Data extraction and quality assessment

The articles were independently screened by both reviewers, who concurred that the selected studies met the inclusion criteria and were of high quality. The risk of bias was assessed using the ROBINS-I tool, designed to evaluate bias in non-randomized studies. This assessment using the ROBINS-I tool was also screened by both reviewers, who reached a consensus on the findings [18].

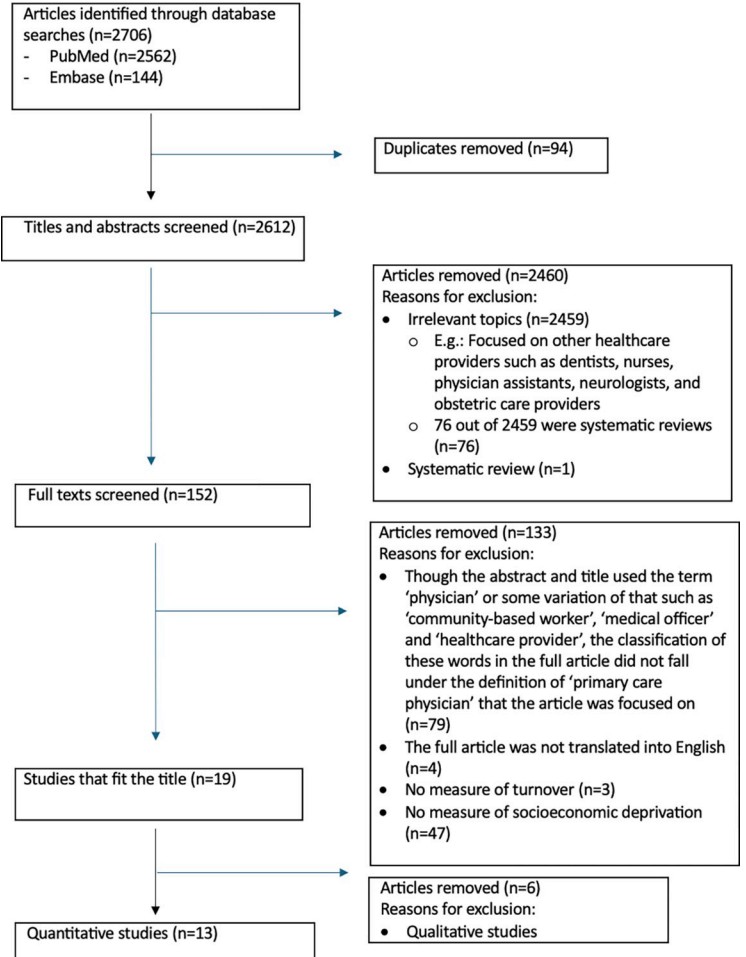

**Fig 1. Flow diagram illustrating the screening process of studies for inclusion in the review.**

### Inclusion criteria

The included studies comprised published cross-sectional, longitudinal, and observational research that featured quantitative assessments of factors influencing primary care physician turnover, or quantitative measurements of the turnover itself. However, the articles included had to have criteria or a method of classification for socio-economically deprived areas. The articles included also focused on quantitative studies. There were no restrictions on the year of the article's publication, though only articles that were either written or translated into English were included. Only studies that focused exclusively on primary care physicians were included and excluded those that examined turnover or retention among wider healthcare staff such as nurses or healthcare assistants. Case studies, review articles, editorials, letters, conferences, abstracts, and opinions were not used.

## Results

After combining the two databases and the search, thirteen quantitative papers were identified and used [8,14,15,19–28]. Eleven out of thirteen papers identified that the level of socioeconomic deprivation in the area of practice was a significant predictor of retention [8,14,15,19–25,27].

These papers were then subsequently grouped on whether they measured turnover or retention, as well as the factors discussed leading to increased turnover. These driving factors were present across both urban and rural areas, though they can be assumed to be more prevalent within areas of deprivation. However, this comparison was not directly measured in the studies included.

## Descriptive statistics and study characteristics

Nine studies measured turnover [8,14,22–28] and four studies measured retention [15,19–21] Amongst the nine studies that measured turnover, there were five observational studies [8,22–25], one cross sectional study [28], two longitudinal studies [26,27] and one comparative study [14]. In comparison, out of the four studies that measured retention, two were observational studies [15,20], one was a cross- sectional study [21] and one was a comparative study [19].

See Table 2 for a full list of study characteristics and measures.

Refer to Table 3 for a list of how the studies quantified socioeconomic deprivation.

For both turnover and retention, observational studies were the most common form of study.

The studies were then categorised into the primary factors highlighted in the article deemed significant for either retaining primary care physicians or influencing turnover. These included: deprivation of the area (n = 7), income (n = 2), burnout (n = 2) and social values (n = 2) (Fig 2).

The chart displays how many articles addressed deprivation, income, burnout and social values as the main contributor to turnover, with deprivation as the most common factor.

The countries reviewed in order of most to least common included England (n = 7), the United States of America (USA) (n = 3), France (n = 2), Scotland (n = 1) and Australia (n = 1). One of the studies was a comparative study that included both Australia and the USA. Therefore, the studies reviewed spanned three continents including Europe, North America and Oceania, with a particular focus on England. This allows for a broader scope of reasons for primary care physician turnover and shows how similar issues transcend borders.

Out of the two studies that measured income as the main factor, both used a form of regression analysis to determine the relationship between income and retention or turnover. Whilst Grigoroglou et al. [22] utilised negative binomial regression and performed descriptive analyses to initially understand the basic distribution of the data, McGrail et al. [23] employed the ordinary least squares regression model. The most frequently utilized measure of deprivation was the Index of Multiple Deprivation, which was employed on five separate occasions [8,21,22,24,28] for papers based in England. One study employed the Maslach Burnout Inventory Human Services survey to quantify burnout which consisted of two items, emotional exhaustion and depersonalisation [28]. Areas of deprivation influencing turnover or retention were analysed using government data/surveys, which included NHS Digital [24], the American Medical Association Physician Masterfile [23], and the Australia Census of Population [14].

## Risk of bias

The risk of bias within the included studies were evaluated using the ROBINS-I tool, as all the studies included were quantitative in nature. Overall, 10 out of the 13 studies had a low risk of bias, with the remaining three studies having a moderate risk of bias. The primary contributors to the moderate risk of bias were confounding factors and missing data. This is depicted visually in Table 4.

The moderate risk of bias in McGrail et al. [14] stemmed from the absence of measurements related to professional factors, such as on-call arrangements, availability of locum relief,

**Table 2. Summary table of results.**

| First author | Year | Country | Design | Sample | Main factor measured | Turnover measure | Retention Measure | Key findings/ Factors influenced | Significant Correlation with turnover |
|---|---|---|---|---|---|---|---|---|---|
| Blane et al. [19] | 2015 | Scotland | Comparative study | 4922 Scottish primary care physicians | Deprivation | | Primary Care Workforce Survey | In deprived areas primary care physician practices tended to be smaller and run single-handedly. Majority of primary care physicians are nearing retirement age. | Yes |
| Chevillard & Mosques [15] | 2021 | France | Observational study | 761 areas with Primary Care Teams vs 1903 areas without to compare the impact of PCTs on GP density | Social values | | Primary Care Team census from the French Ministry of Health | Primary Care Teams (PCTs) have varying impacts on attracting and retaining GPs. In suburban regions, PCTs have a positive effect. However, in rural areas, there is no substantial increase in attraction or retention. | Yes |
| Chevillard et al. [20] | 2019 | France | Observational study | 1416 rural living areas: 184 areas with PCTs and 1,232 areas without PCTs. | Social values | | Primary Care Team census from the French Ministry of Health | Primary Care Teams attract and retain GPs, especially in deprived areas, with an increase of 3.5 GPs per 100,000 inhabitants compared to similar areas without PCTs | Yes |
| Ding et al. [21] | 2008 | England | Serial cross-sectional study | All GPs practising in England during the years 1996/97, 2000/01, 2004/05 | Burnout | | GMS and PMS data (GP census) available on The Information Centre: annually records all practising GPs in England | Salaried GPs in the UK are more likely to be female and overseas trained, mainly seeking flexibility and part time work. This has allowed previously underutilised GPs to participate in patient care. Whilst these flexible roles were initially introduced to attract and retain GPs in deprived areas, they are not more common in affluent areas. | Yes |
| Grigoroglou et al. [22] | 2022 | England | Observational study | 1217 locum primary care physician FTE | Income | UK Quality and Outcomes Framework | | Increased locum use in primary care physician practices within rural areas. | Yes |
| McGrail et al. [23] | 2017 | United States | Observational study | 1.4 million location pairings of primary care physicians | Deprivation | American Medical Association Physician Masterfile | | Increased turnover of physicians correlated with deprived areas partially due to lack of hospital resources. | Yes |

*(Continued)*

**Table 2.** (Continued)

| First author | Year | Country | Design | Sample | Main factor measured | Turnover measure | Retention Measure | Key findings/ Factors influenced | Significant Correlation with turnover |
|---|---|---|---|---|---|---|---|---|---|
| McGrail et al. [14] | 2017 | Australia and the United States | Comparative study | Exact number not specified. Provider to population ratios in 1949 USA rural counties and 370 Australian rural areas | Income | American community Survey and Australia Census of Population | | Australia has a higher supply of primary care physicians in older and lower income areas compared to the USA. Increase in income did not improve retention in smaller and poorer towns. | Yes |
| Parisi et al. [24] | 2021 | England | Retrospective study of cross-sectional data | 8085 primary care physician Practices in 2007 6598 in 2019 | Deprivation | NHS digital | | Increase in primary care physician turnover within last decade, especially within more deprived areas. | Yes |
| Parisi et al. [8] | 2023 | England | Retrospective observational study | Average of 7526 practices per year | Deprivation | primary care physician Patient survey from 2007–2019 | | Higher primary care physician turnover was associated with a deprived area of practice and larger practice size. | Yes |
| Taylor et al. [25] | 1998 | England | Observational Study | 2922 primary care physicians | Deprivation | General Practitioner Census from NHS executive | | Turnover is positively associated with areas of high need. | Yes |
| Taylor et al. [26] | 1999 | England | Longitudinal study | 252 GPs that fit the criteria: 1. GP was a new entrant into general practice 2. 35 years old or less | Deprivation | General Practitioner Census from NHS consecutive | | Deprivation of the area of the practice was found to not be a significant predictor of retention. Deprivation when measured in terms of the GP's patients had a slight positive association with retention. Instead, there was a reduction in turnover rates in larger practices. | No |
| Vanasse et al. [27] | 2007 | United States | Longitudinal study (examines data from 1981 and 2003) | 83 383 primary care physicians | Deprivation | American Medical Association Masterfiles | | Though deprivation was a factor, there was a stronger association with increased migration out of areas with a greater proportion of ethnic minorities. | Yes |
| Zhou et al. [28] | 2022 | England | Cross-sectional survey | 387 primary care physicians | Burnout | 10 item online questionnaire on primary care physician characteristics. For burnout, Maslach Burnout inventory was used. | | Primary care physicians who reported greater diagnostic uncertainty had increased emotional exhaustion, job dissatisfaction and increased turnover intention. | No |

**Table 3. Summary table detailing how socioeconomic deprivation was measured for each article.**

| Article Author | How was socioeconomic deprivation measured? |
| --- | --- |
| Blane et al. [19] | Scottish Index of Multiple Deprivation (SIMD): practices were divided into deciles based on the percentage of patients in the top 15% of the most deprived postcodes weighted by population and overall SIMD score. |
| Chevillard & Mosques [15] | Living areas classified into 6 sub-groups; 2 out of which were medically underserved. One group was characterised by a younger population with a higher amount of blue-collar workers and average access to healthcare. The second with rural and deprived areas with a greater proportion of elderly and blue-collar workers |
| Chevillard et al. [20] | Cluster 1: suburban areas with a younger population underserved by GPs and nurses despite good socio-economic status (education, occupation and income). Cluster 2: increased socio-economic status areas with an average level of accessibility of primary health care professionals. Cluster 3: industrial and agricultural areas with lower accessibility of primary health care professionals Cluster 4: low socio-economic status populations with lower accessibility of primary health care professionals. Cluster 5: tourist areas with greatest accessibility of primary health care Cluster 6: remote areas with an older population. Accessibility of primary care is close to the average, but the GPs are older |
| Ding et al. [21] | 2004 Index of Multiple Deprivation |
| Grigoroglou et al. [22] | Deprivation scores from the Index of Multiple Deprivation (IMD) were assigned to practices based on their postcode. The IMD is a measure of deprivation for the 32,844 small areas: Lower Layer Super Output Area (LSOAs) in England. Each area was given a score between 0 and 100, with higher scores indicating higher levels of deprivation. |
| McGrail et al. [23] | Practices were classified according to the 9 level Rural Urban Continuum Codes and combined with 3 population groupings (<2500, 2500–20000, >20000) and adjacency to metropolitan area. Deprivation was calculated using household income and median house prices for reference. |
| McGrail et al. [14] | Provider to population ratios calculated. Deprivation was also measured using US Medically Underserved Areas and Health Professional Shortage Areas and Australia's District of Workforce Shortage Areas and Areas of Need |
| Parisi et al. [24] | Index of Multiple Deprivation categorised in quintiles |
| Parisi et al. [8] | Index of Multiple Deprivation and size of practice population was calculated in quintiles |
| Taylor et al. [25] | Used deprivation bands 1–3. Also used Hacking's measure of relative under/overprovision of GP services weighted for need |
| Taylor et al. [26] | Deprivation was measured using the health authority deprivation score (UPA91) and the proportion of GP's patients that were on deprivation payments Health authority level deprivation was measured by the Jarman index, also known as the Underprivileged Area score, which takes into account unemployment, overcrowded housing, single parent households, elderly, children under five, ethnic minorities, poverty and residential mobility. |
| Vanasse et al. [27] | Deprivation was measured via the mean per capita income of the population, (the ratio of hospital beds per 1000 population) and the rural or urban status of the geographic location |
| Zhou et al. [28] | Deprivation was calculated using the Index of Multiple Deprivation quintiles which included the seven domains of: income, employment, health, living environment, barriers to housing, services and crime rates |

and infrastructure quality. This lack of data made it difficult to determine whether these factors outweighed community amenities in influencing primary care physicians' decisions on practice location.

Parisi et al. [24] and Parisi et al. [8] were classified as having a moderate risk of bias due to the limitation of the covariates available within the national administrative datasets, and

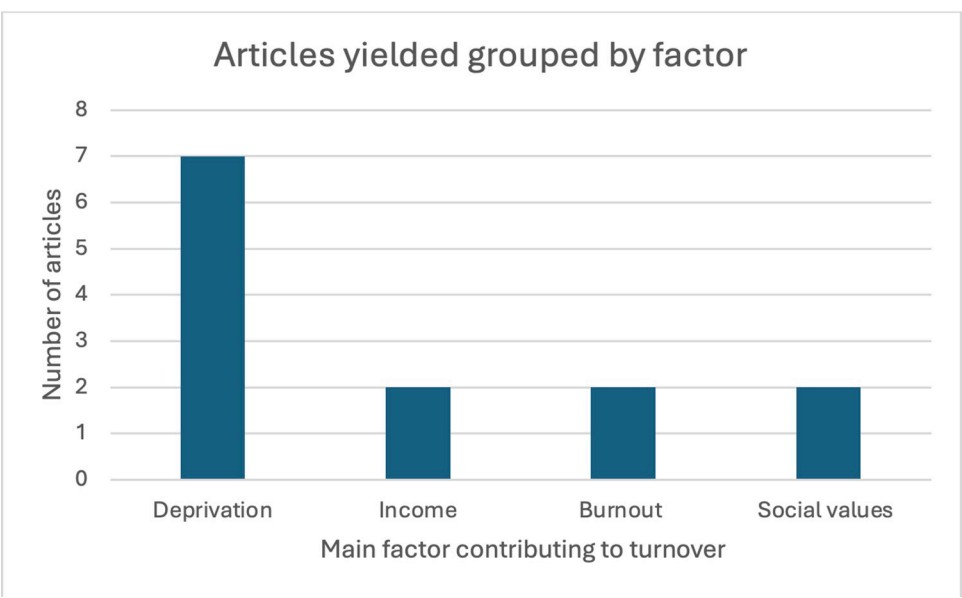

**Fig 2. Column chart illustrating the grouping of articles based on the main factors contributing to turnover.**

therefore were unable to cover all factors relevant to turnover. Therefore, this has the possibility of missing important variables that could potentially skew the results.

Overall, most studies included adequately addressed the risk of bias within their data collection, with any remaining risk being either minimal or effectively mitigated.

## Income findings

Two studies measured income influencing turnover and retention [22,23]. None of the two studies had deemed income as an important determinant for the retention of primary care physicians.

Furthermore, McGrail et al. [14] compared Australia and the US, arguing that an increase in income did not encourage doctors to work in areas that had a shortage of primary care physicians. Although both countries had similar geographical characteristics; in Australia, the level of remoteness was strongly correlated with a poorer supply. Although rurality does not always directly correlate with socioeconomic deprivation, rural practices often face significant challenges, including limited accessibility for patients and a shortage of available healthcare facilities. As a result, rural areas frequently become medically underserved [14]. Despite the lower housing costs within these areas, doctors are amongst the top income earners and are therefore willing to pay for more affluent neighbourhoods. Instead, turnover was primarily driven by insufficient support and inadequate time off [10], with factors contributing to primary care physician retention encompassing larger population sizes, and the overall affluence and safety of the area, important considerations for individuals raising families [14].

There is also the influence of each country's respective medical insurance systems. In Australia, there is greater availability of primary care physicians in communities with older populations characterised by higher rates of unemployment and lower income compared to the USA, where there is a shortage in areas with more uninsured residents. This is due to Australia's universal healthcare system, known as 'Medicare,' which helps mitigate the decline in doctor availability in poorer areas, resulting in a less pronounced decrease in supply compared to the USA.

**Table 4. Visual depiction of ROBINS-I bias assessment.**

| Author/ Year | Bias due to confounding | Bias in selection of participants into the study | Bias due to deviations from intended interventions | Bias due to missing data | Bias in measurement of outcomes | Bias in selection of the reported result | Overall bias |
|---|---|---|---|---|---|---|---|
| Blane 2015 [19] | Low risk of bias | Low risk of bias | Low risk of bias | Low risk of bias | Low risk of bias | Low risk of bias | Low risk of bias |
| Chevillard 2021 [15] | Low risk of bias | Low risk of bias | Low risk of bias | Low risk of bias | Low risk of bias | Low risk of bias | Low risk of bias |
| Chevillard 2019 [20] | Low risk of bias | Low risk of bias | Low risk of bias | Low risk of bias | Low risk of bias | Low risk of bias | Low risk of bias |
| Ding 2008 [21] | Low risk of bias | Low risk of bias | Low risk of bias | Low risk of bias | Low risk of bias | Low risk of bias | Low risk of bias |
| Grigoroglou 2022 [22] | Low risk of bias | Low risk of bias | Low risk of bias | Low risk of bias | Low risk of bias | Low risk of bias | Low risk of bias |
| McGrail 2017 [23] | Low risk of bias | Low risk of bias | Low risk of bias | Low risk of bias | Low risk of bias | Low risk of bias | Low risk of bias |
| McGrail 2017 [14] | Low risk of bias | Low risk of bias | Low risk of bias | Moderate risk of bias | Low risk of bias | Low risk of bias | Moderate risk of bias |
| Parisi 2021 [24] | Moderate risk of bias | Low risk of bias | Low risk of bias | Moderate risk of bias | Low risk of bias | Low risk of bias | Moderate risk of bias |
| Parisi 2023 [8] | Moderate risk of bias | Low risk of bias | Low risk of bias | Moderate risk of bias | Low risk of bias | Low risk of bias | Moderate risk of bias |
| Taylor 1998 [25] | Low risk of bias | Low risk of bias | Low risk of bias | Low risk of bias | Low risk of bias | Low risk of bias | Low risk of bias |
| Taylor 1999 [26] | Low risk of bias | Low risk of bias | Low risk of bias | Low risk of bias | Low risk of bias | Low risk of bias | Low risk of bias |
| Vanasse 2007 [27] | Low risk of bias | Low risk of bias | Low risk of bias | Low risk of bias | Low risk of bias | Low risk of bias | Low risk of bias |
| Zhou 2022 [28] | Low risk of bias | Low risk of bias | Low risk of bias | Low risk of bias | Low risk of bias | Low risk of bias | Low risk of bias |

Locums have been used to relieve primary care physician shortages in disadvantaged or socio-economically deprived areas. Traditionally, locums have been used to temporarily cover vacancies or provide relief for permanent staff on leave. However, one of the studies in England reflect the modified role of locums as a longer-term solution to cover the staffing shortage, especially within rural regions, representing the wider issue of recruitment and retention of primary care physicians within these areas [22]. Therefore, income is not a significant factor in retaining primary care physicians, with community satisfaction, professional opportunities and support systems playing a more crucial role.

## Burnout findings

Two studies [21,28] agreed burnout was the main factor influencing retention and turnover of primary care physicians.

Zhou et al. [28] did not find an association between socioeconomic deprivation and burnout, measured by emotional exhaustion and depersonalisation. Instead, diagnostic uncertainty, even amongst experienced GPs and sickness presenteeism had a stronger link to burnout and hence GP turnover. Furthermore, full-time GPs were more likely to burnout, which suggests organisational changes across the board need to be made to reduce workload and improve support to ultimately improve GP retention.

Ding et al. [21] emphasises this point and found that the position of salaried GPs which includes part time work and more flexible hours have reduced burnout. The position of salaried GPs has allowed previously underutilised GPs such as women of child-bearing age, newly

qualified GPs or those nearing retirement, to contribute more effectively to patient care. Despite the increase in mobility due to shorter- term contracts, the position of salaried GPs are more concentrated in affluent areas, though there is an increase in GP workforce retention in deprived areas, albeit at a lower rate than affluent areas.

## Deprivation findings

Seven studies [8,19,23–27] focused on the impact of socio-economic status of the area of practice itself as the significant factor that contributed to high turnover and low retention rates. Refer to Table 3 to determine how the level of socio-economic deprivation was measured in each study. Blane et al. [19] showed how in Scotland, despite more practices in the more economically deprived deciles, there were fewer primary care physicians available. Furthermore, these areas had a higher overall proportion of older primary care physicians, with over a third aged 50 or older and nearing retirement. This highlights the inverse care law, with a lack of healthcare available for the people who need it the most. This trend has been evident in Taylor et al. [25] from 1990–1994 England, quantifying primary care physician turnover and migration where the relatively deprived areas lost 23 primary care physicians compared with the relatively over-provided areas that gained 3 primary care physicians over the study period.

McGrail et al. [23] highlighted the increased mobility in remote USA areas, as they lacked professional support and hospital resources. This perpetuates the issue within these rural that are often disadvantaged, as increased turnover increases strain on the already finite resources and staff, which would further discourage primary care physician retention. This is furthered by Parisi et al. [24], where the most deprived areas had a higher likelihood of GP turnover compared with practices located in the least deprived areas in England.

However, Taylor et al. [26] reviewed retention rates in 1991–2 England, and suggested an initial high retention rate of 85% for young new entrant GPs. This contrasted sharply with the 82% rate of intention to leave observed among GPs in England in 2015 [9]. The Taylor et al. [26] 1991–92 study noted the large impact that a loss of a small number of GPs had on local health supply, with larger GP practices reducing the likelihood of younger GPs working. The proportion of patients who were classified as having a poorer socio-economic background did not negatively affect retention, and in some cases, improved retention rates. The study also found that working in an area that is overall deprived did not significantly reduce the likelihood of those GPs staying in their jobs. Furthermore, unlike the findings of Vanasse et al. [27], this research indicated the proportion of ethnic minorities did not appear to affect retention rates. This discrepancy may be attributed by the context of the USA where Vanasse et al. [27] was conducted, as it identified a potential link between poverty and a higher proportion of ethnic minority populations, which could influence retention.

## Social values findings

A review of Primary Care Teams (PCTs) in France examines their impact on the retention of GPs across different socioeconomic regions in France [15,20]. The trend of promoting group practice in primary care with the development of Primary Care Teams (PCTs) where GPs are working collaboratively with other healthcare professionals such as midwives, dentists, paramedics, nurses and other administrative staff have not only increased collaboration and reduced professional isolation but have also improved working conditions contributing to a better work life balance.

Chevillard & Mousques [15] highlights the role of PCTs in addressing geographical inequalities of GPs by reducing disparities in underserved areas. It suggests that group practices such as PCTs are more effective in attracting GPs compared to the traditional incentive of financial

bonuses. Similarly, Chevillard et al. [20] comments within deprived rural regions, the implementation of PCTs have helped gain an average of 3.5 GPs per 100,000, improving healthcare accessibility.

## Discussion

The identified studies found elevated turnover amongst primary care physicians stemmed from factors such as burnout and a lack of resources, rather than income, which was the traditional method of attracting primary care physicians to work in medically underserved areas [29]. The findings of this systematic review included studies based on high income countries with universal health coverage, where the Inverse Care Law is described as 'incomplete' despite the positive correlation of healthcare expenditure to social disadvantage, as they still receive less healthcare than they need, given their greater health challenges [5]. This differs in low to middle income countries, where there is a 'complete' inverse care law where there is a reduction in healthcare utilisation associated with socioeconomic disadvantage. This occurs as the allocation of healthcare resources and spending per capita remains inversely related to social disadvantage with the prioritisation of economic or political power over actual health needs [5].

The articles included in the search based in Scotland, Australia and the USA [14,19] found a significant correlation between socio-economic deprivation and primary care physician turnover, characterised by additional challenges unrelated to working conditions that contribute to high turnover rates. These included a lack of facilities or communal areas, employment opportunities for their partners and families, and safety concerns due to higher rates of crime in these lower income areas [14]. Therefore, investment is needed to enhance the overall appeal of these towns, which will also indirectly ease burnout rates in rural areas with the recruitment of more long-term primary care physicians.

In contrast, studies conducted in England [21,28] identified working hours as a more significant factor influencing physician turnover. This difference may be attributed to the differing geographical and socioeconomic contexts of the countries examined. For instance, Australia and the USA are geographically larger, and along with Scotland, may experience greater variability in levels of socioeconomic deprivation, though a direct comparison of these factors was not explicitly measured. Therefore, policymakers must consider the unique geographical characteristics of each country to develop tailored strategies that effectively reduce primary care physician turnover.

The success of Primary Care Teams (PCTs) in attracting GPs to underserved areas suggests that governments should prioritize fostering collaboration between GPs and allied healthcare professionals, rather than relying solely on financial incentives [15]. By reducing professional isolation and creating more cooperative working environments, PCTs can help alleviate burnout and improve working conditions. Overall, these findings challenge the previous assumption that income is the most significant factor in physician retention. Instead, factors like community satisfaction, professional opportunities, support, and burnout are critical to improving retention. While socioeconomic deprivation remains a challenge, models like PCTs and more flexible working arrangements show promise in addressing these issues, especially in underserved areas.

### Strengths and limitations

This systematic review synthesised current research and provided a global, overarching view of the factors that lead to high primary care physician turnover, especially in the areas of deprivation. This study also investigated the factors that contributed to the retention of primary

care physicians in these areas, providing valuable insights for future policymakers to tailor their strategies accordingly.

Despite the large geographical area covered, there is room for more literature to be covered on the rest of the world, especially in developing countries. This approach would allow for a broader range of turnover factors to be identified, enabling the discovery of any commonalities among them. This systematic review is restricted by the exclusion of papers that were not in English, as well as articles where the full text was not available. Despite the vast majority of articles covered on PubMed and Embase, the decision to exclude other databases and the lack of access to an information scientist may have also slightly limited the breadth of this review. Furthermore, while this study concentrated on the primary factor discussed in each article, some articles mentioned additional factors contributing to turnover, resulting in some overlap. Despite this, all the main factors were thoroughly documented in this systematic review. Lastly, due to the nature of the question being focused on the factors that arise from socio-economic deprivation, this review may not have been able to identify all the potential influencing factors of primary care physician turnover.

## Implications for practice and future work

Current studies show how the main methods for retention have been the employment of locums within the UK [22], along with incentivisation of an increased salary in the UK, Australia and Canada [10–12]. However, this systematic review suggests a more holistic and comprehensive approach is needed to improve primary healthcare practices. Efforts of enhancing infrastructure, developing community facilities, creating more job opportunities, improving safety and providing better educational and recreational options, can make these areas more appealing to potential residents, and help retain the primary care physicians already living there. Furthermore, the option of flexible work arrangements such as telephone consultations or flexible scheduling such as part time work may also incentivise the long-term retention of primary care physicians.

For future research, the impact of various interventions needs to be investigated through long-term and longitudinal studies. This includes examining the effects of investment in deprived regions, and the implementation of primary care teams or equivalent to foster a collaborative working environment and reduce work isolation. Furthermore, research into the role of cultural competence training can enhance primary care physicians' ability to work in diverse and socioeconomically deprived communities. This will better equip primary care physicians to provide sensitive, effective and equitable care to all patients, regardless of background or circumstance.

## Conclusion

The poor retention and high turnover rate of primary care physicians, especially within socio-economically deprived regions transcends geographical borders. This concerns patient safety due to the disruption of continuity of care, leading to increased hospital admissions and thus poorer outcomes. Therefore, by utilising current research that highlight the issues within these areas, future government policies are able to focus on the retention of doctors within these disadvantaged regions.

## Supporting information

**S1 Table. PRISMA checklist.**
(DOCX)

**S2 Table. Reasons for exclusion of study.**
(XLSX)

**S1 File. Extraction of search strategy.**
(DOCX)

## Author Contributions

**Conceptualization:** Evangelos Kontopantelis.

**Data curation:** Jasmine Lee.

**Formal analysis:** Jasmine Lee.

**Investigation:** Jasmine Lee.

**Methodology:** Jasmine Lee.

**Project administration:** Evangelos Kontopantelis.

**Supervision:** Evangelos Kontopantelis.

**Validation:** Evangelos Kontopantelis.

**Visualization:** Jasmine Lee.

**Writing – original draft:** Jasmine Lee.

**Writing – review & editing:** Jasmine Lee, Evangelos Kontopantelis.

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
