## [Decision Letter · Decision Letter 0]

3 Sep 2024

PONE-D-24-36570A systematic review exploring the factors that contribute to increased primary care physician turnover in socio-economically deprived areas.PLOS ONE

Dear Dr. Lee,

Thank you for submitting your manuscript to PLOS ONE. After careful consideration, we feel that it has merit but does not fully meet PLOS ONE’s publication criteria as it currently stands. Therefore, we invite you to submit a revised version of the manuscript that addresses the points raised during the review process. The reviewers have both noted the importance of this topic and have provided recommendation for improvements based on inclusion &/or strengthening of contextual elements of the work.

We look forward to receiving your revised manuscript.

Kind regards,

Jenny Wilkinson, PhD

Academic Editor

PLOS ONE

Journal Requirements:

4. As required by our policy on Data Availability, please ensure your manuscript or supplementary information includes the following: 

Reviewers' comments:

Reviewer's Responses to Questions

**Comments to the Author**

1. Is the manuscript technically sound, and do the data support the conclusions?

Reviewer #1: Yes

Reviewer #2: Partly

2. Has the statistical analysis been performed appropriately and rigorously? 

Reviewer #1: N/A

Reviewer #2: N/A

3. Have the authors made all data underlying the findings in their manuscript fully available?

Reviewer #1: Yes

Reviewer #2: Yes

4. Is the manuscript presented in an intelligible fashion and written in standard English?

Reviewer #1: Yes

Reviewer #2: Yes

5. Review Comments to the Author

Reviewer #1: This is a well conducted review on a very important topic, and as such is worthy of publication in this journal. With regards the issues facing GPs in deprived areas, its is essential that the authors situate this within the inverse care law, first described by Todor-Hart in 1971 but equally relevant today. Mercer and Watt (2007) in Annals of Family Medicine were the first to show how the inverse care law operates in primary care in deprived areas, and recent work in Scotland has identified that little has changed, see; Health Foundation report 2024, Tackling the inverse care law in Scottish primary care. Similar recent work by the Health Foundation has shown that the inverse care law is still an issue, and the Lancet series on the inverse care law in 2021 showed that its a global problem. Thus the burn-out and turnover of GPs in deprived areas is not just because patients have more complex problems but crucially because supply does not match need. There needs to be a radical increase in GP funding together with a redistribution of funding according to deprivation. This information needs to be added to the Introduction, and brought into the discussion. I found the discussion odd as it had very little about comparison with previous studies, and a lot about strengths and limitations, and under policy, no mention whatsoever of the inverse care law. This is a glaring omission. So I suggest the Introduction and Discussion need to be substantially revised.

Reviewer #2: Many thanks for the chance to review this article. My main concerns are that the aim is not clearly articulated at the start and the article itself is confusing at times. I am not clear what the authors mean by socioeconomic deprivation (SED), and I am not clear how this was objectively measured in the different studies identified. In addition, there is no clear discussion how rurality influences this – and several times the paper seems to equate issues with rurality with socio-economic deprivation. It would be supposed that deprivation in rural and urban areas differs and so some discussion on how it has been measured in both these contexts and an exploration of whether rurality made any difference (hard to see how it wouldn’t) is important. In addition, I am concerned that this is a mixed synthesis but the approach to qualitative findings misses the purpose of qualitative work completely, not least in how it approaches data quality assessment. Finally, the significant exclusion of articles on the basis of title screening alone is a bit of a concern for me. Please see attachment for more details

6. PLOS authors have the option to publish the peer review history of their article (what does this mean?). If published, this will include your full peer review and any attached files.

Reviewer #1: No

Reviewer #2: **Yes: **Marianne McCallum

---

## [Author Response · Author response to Decision Letter 0]

28 Sep 2024

We would like to thank the reviewers and the editor for their valuable feedback. We provide a point-by-point response below to each of the reviewers’ comments. 

Reviewer 1 comments: 

This is a well conducted review on a very important topic, and as such is worthy of publication in this journal. With regards the issues facing GPs in deprived areas, its is essential that the authors situate this within the inverse care law, first described by Todor-Hart in 1971 but equally relevant today. Mercer and Watt (2007) in Annals of Family Medicine were the first to show how the inverse care law operates in primary care in deprived areas, and recent work in Scotland has identified that little has changed, see; Health Foundation report 2024, Tackling the inverse care law in Scottish primary care. Similar recent work by the Health Foundation has shown that the inverse care law is still an issue, and the Lancet series on the inverse care law in 2021 showed that its a global problem. Thus the burn-out and turnover of GPs in deprived areas is not just because patients have more complex problems but crucially because supply does not match need. There needs to be a radical increase in GP funding together with a redistribution of funding according to deprivation. This information needs to be added to the Introduction, and brought into the discussion. I found the discussion odd as it had very little about comparison with previous studies, and a lot about strengths and limitations, and under policy, no mention whatsoever of the inverse care law. This is a glaring omission. So I suggest the Introduction and Discussion need to be substantially revised.

Response:

Thank you for your kind assessment. We have now revised the introduction to include Inverse Care Law (See page 4 paragraph 1):

“This aligns with the Inverse Care law, which suggests the population who are most in need of healthcare services are often the least likely to receive them as supply is unable to keep up with need (4). This trend has continued from its initial establishment of this issue in 1971 till today and is observed internationally, not only in low or middle income countries, but also in countries classified as high income such as the UK and the USA which the included studies within this systematic review focuses on (5). Therefore, there needs to have a significant boost in GP funding, coupled with a more equitable distribution of resources that prioritises areas with higher levels of deprivation.”

 and also the discussion to include previous studies on the Inverse Care Law (see page 31 paragraph 1).

“The findings of this systematic review included studies based on high income countries with universal health coverage, where the Inverse Care Law is described as ‘incomplete’ despite the positive correlation of healthcare expenditure to social disadvantage, as they still receive less healthcare than they need, given their greater health challenges.(5) This differs in low to middle income countries, where there is a ‘complete’ inverse care law where there is a reduction in healthcare utilisation associated with socioeconomic disadvantage. This occurs as the allocation of healthcare resources and spending per capita remains inversely related to social disadvantage with the prioritisation of economic or political power over actual health needs.(5)”

Reviewer 2 comments: 

Many thanks for the chance to review this article. My main concerns are that the aim is not clearly articulated at the start and the article itself is confusing at times. I am not clear what the authors mean by socioeconomic deprivation (SED)

Response:

Thank you, we have now clarified this (page 4 paragraph 1 line 3) 

“Socioeconomic deprivation was broadly defined based on the parameters of the Index of Multiple Deprivation Score which takes into consideration employment, education and training, health, housing, and living environment deprivation.”

Comment: 

and am not clear how this was objectively measured in the different studies identified. 

Response:

Thank you, we have now created a table detailing how SED was objectively measured for each article- See page 20-22 

Article Author How was socioeconomic deprivation measured?

Blane 2015 Scottish Index of Multiple Deprivation (SIMD): practices were divided into deciles based on the percentage of patients in the top 15% of the most deprived postcodes weighted by population and overall SIMD score. 

Chevillard 2021 Living areas classified into 6 sub-groups; 2 out of which were medically underserved. 

One group was characterised by a younger population with a higher amount of blue-collar workers and average access to healthcare. 

The second with rural and deprived areas with a greater proportion of elderly and blue-collar workers 

Chevillard 2019 Cluster 1: suburban areas with a younger population underserved by GPs and nurses despite good socio-economic status (education, occupation and income). 

Cluster 2: increased socio-economic status areas with an average level of accessibility of primary health care professionals. 

Cluster 3: industrial and agricultural areas with lower accessibility of primary health care professionals 

Cluster 4: low socio-economic status populations with lower accessibility of primary health care professionals.

Cluster 5: tourist areas with greatest accessibility of primary health care 

Cluster 6: remote areas with an older population. Accessibility of primary care is close to the average, but the GPs are older 

Ding 2008 2004 Index of Multiple Deprivation 

Grigoroglou 2022 Deprivation scores from the Index of Multiple Deprivation (IMD) were assigned to practices based on their postcode. The IMD is a measure of deprivation for the 32,844 small areas: Lower Layer Super Output Area (LSOAs) in England. Each area was given a score between 0 and 100, with higher scores indicating higher levels of deprivation.

McGrail 2017: Mobility of US Rural Primary Care Physicians during 2000-2014 Practices were classified according to the 9 level Rural Urban Continuum Codes and combined with 3 population groupings (<2500, 2500-20000, >20000) and adjacency to metropolitan area.

Deprivation was calculated using household income and median house prices for reference.

McGrail 2017: Measuring the attractiveness of rural communities in accounting for differences of rural primary care workforce supply Provider to population ratios calculated. 

Deprivation was also measured using US Medically Underserved Areas and Health Professional Shortage Areas and Australia’s District of Workforce Shortage Areas and Areas of Need 

Parisi 2021 Index of Multiple Deprivation categorised in quintiles 

Parisi 2023 Index of Multiple Deprivation and size of practice population was calculated in quintiles 

Taylor 1998 Used deprivation bands 1-3. Also used Hacking’s measure of relative under/overprovision of GP services weighted for need 

Taylor 1999 Deprivation was measured using the health authority deprivation score (UPA91) and the proportion of GP’s patients that were on deprivation payments 

(Health authority level deprivation was measured by the Jarman index, also known as the Underprivileged Area score, which takes into account unemployment, overcrowded housing, single parent households, elderly, children under five, ethnic minorities, poverty and residential mobility.)

Vanasse 2007 Deprivation was measured via the mean per capita income of the population, (the ratio of hospital beds per 1000 population) and the rural or urban status of the geographic location 

Zhou 2022 Deprivation was calculated using the Index of Multiple Deprivation quintiles which included the seven domains of: income, employment, health, living environment, barriers to housing, services and crime rates

Comment:

In addition, there is no clear discussion how rurality influences this – and several times the paper seems to equate issues with rurality with socio-economic deprivation. It would be supposed that deprivation in rural and urban areas differs and so some discussion on how it has been measured in both these contexts and an exploration of whether rurality made any difference (hard to see how it wouldn’t) is important. 

Response:

Thank you for your feedback. We have now clarified this point by distinguishing rurality and socio-economic deprivation on page 27 on the second paragraph:

“Although rurality does not always directly correlate with socioeconomic deprivation, rural practices often face significant challenges, including limited accessibility for patients and a shortage of available healthcare facilities. As a result, rural areas frequently become medically underserved.(40)”

Comment: 

In addition, I am concerned that this is a mixed synthesis but the approach to qualitative findings misses the purpose of qualitative work completely, not least in how it approaches data quality assessment.

Response: 

Thank you for your feedback. We have now revised the inclusion criteria on page 9 to now only include quantitative studies. 

See methods on page 2: 

“The eligibility criteria included quantitative empirical studies that included a measurement of at least one of the factors behind increased primary care physician turnover or retention within socio-economically deprived or disadvantaged areas. However, the included studies were required to employ a specific methodology for classifying or defining socioeconomic deprivation.”

Also, see the revised inclusion criteria on page 9: 

“The included studies comprised published cross-sectional, longitudinal, and observational research that featured quantitative assessments of factors influencing primary care physician turnover, or quantitative measurements of the turnover itself. However, the articles included had to have criteria or a method of classification for socio-economically deprived areas. The articles included also focused on quantitative studies.”

Comment: 

Finally, the significant exclusion of articles on the basis of title screening alone is a bit of a concern for me. 

Response: 

Thank you for your feedback. We have addressed this concern by changing the initial screening process in our updated search. The first step in the screening process is now based on title and abstract. See page 7 paragraph 2 under study selection. 

“A flow chart demonstrating the selection process is presented in Figure 1. After the removal of the duplicate articles, titles and abstracts were screened and filtered. This step was performed twice to ensure no articles were missed before a third filtering of the full text was performed. Both reviewers are in agreement regarding the selection of the articles included in the study.”

Comment:

Major points:

The aim of this review is not clear and this is critical. The background suggests turnover is known to be higher in areas of high SED and yet part of the aim states the purpose of this review is to see if this is the case (of note if this was looked at it is not clearly shown in the results section). In the aim paragraph it would make more sense to me if it started with the second sentence which SEEMS to be the aim of this review – to look at what factors influence retention and turnover in high SED areas. I would also suggest leaving the second aim out as I am not sure the search is really set up to answer this question? 

Response:

Thank you for the suggestion. We have taken on board the feedback and removed the second aim. 

Furthermore, the aim has now been revised and clarified as requested: see page 5 paragraph 3 for revised aim: 

“Therefore, this study aims to synthesise existing research on the factors contributing to primary care physician turnover and retention within socio-economically deprived areas to offer a coherent, global perspective. This in turn allows the examination of how these specific gaps can be amended by policy interventions, although we did not explicitly search for these.”

Comment: 

In addition, the aim doesn’t entirely make clear what the authors mean by SED and at times it seems that this review is discussing medically underserved areas either due to SED or rurality. Critically there is no discussion about how the included studies measured SED. 

Response: 

Thank you for your feedback. We have made the following corrections to distinguish between SED and rurality: 

Page 5 paragraph 1 line 7: correction: the scheme was specifically for areas of doctors, which were often the poorest areas. The phrase “rural or deprived” was removed. 

Similarly, the introduction of the Targeted Enhanced Recruitment Scheme in the UK offers primary care physician trainees a £20,000 payment to train in areas deprived of doctors in Scotland, which are often also socio-economically deprived (14). However, only 21% of doctors reported being influenced by the program (13).

The distinction between rurality and deprivation was also addressed directly in page 27 paragraph 1. 

“Although rurality does not always directly correlate with socioeconomic deprivation, rural practices often face significant challenges, including limited accessibility for patients and a shortage of available healthcare facilities. As a result, rural areas frequently become medically underserved.(40)”

Comment: 

Several of the studies appear to be large surveys not just targeting high SED areas and it is not clear how the authors ensured they extracted the data that related to high SED, 

Response: 

Thank you for your feedback. We have clarified this now as the criteria for the search was changed and the articles included now specifically target high SED areas. See inclusion criteria on page 9 paragraph 2: 

“The included studies comprised published cross-sectional, longitudinal, and observational research that featured quantitative assessments of factors influencing primary care physician turnover, or quantitative measurements of the turnover itself. However, the articles included had to have criteria or a method of classification for socio-economically deprived areas. The articles included also focused on quantitative studies:

Furthermore, the table from page 20-22 details how each article measured SED 

Article Author How was socioeconomic deprivation measured?

Blane 2015 Scottish Index of Multiple Deprivation (SIMD): practices were divided into deciles based on the percentage of patients in the top 15% of the most deprived postcodes weighted by population and overall SIMD score. 

Chevillard 2021 Living areas classified into 6 sub-groups; 2 out of which were medically underserved. 

One group was characterised by a younger population with a higher amount of blue-collar workers and average access to healthcare. 

The second with rural and deprived areas with a greater proportion of elderly and blue-collar workers 

Chevillard 2019 Cluster 1: suburban areas with a younger population underserved by GPs and nurses despite good socio-economic status (education, occupation and income). 

Cluster 2: increased socio-economic status areas with an average level of accessibility of primary health care professionals. 

Cluster 3: industrial and agricultural areas with lower accessibility of primary health care professionals 

Cluster 4: low socio-economic status populations with lower accessibility of primary health care professionals.

Cluster 5: tourist areas with greatest accessibility of primary health care 

Cluster 6: remote areas with an older population. Accessibility of primary care is close to the average, but the GPs are older 

Ding 2008 2004 Index of Multiple Deprivation 

Grigoroglou 2022 Deprivation scores from the Index of Multiple Deprivation (IMD) were assigned to practices based on their postcode. The IMD is a measure of deprivation for the 32,844 small areas: Lower Layer Super Output Area (LSOAs) in England. Each area was given a score between 0 and 100, with higher scores indicating higher levels of deprivation.

McGrail 2017: Mobility of US Rural Primary Care Physicians during 2000-2014 Practices were classified according to the 9 level Rural Urban Continuum Codes and combined with 3 population groupings (<2500, 2500-20000, >20000) and adjacency to metropolitan area.

Deprivation was calculated using household income and median house prices for reference.

McGrail 2017: Measuring the a

---

## [Decision Letter · Decision Letter 1]

18 Oct 2024

PONE-D-24-36570R1A systematic review exploring the factors that contribute to increased primary care physician turnover in socio-economically deprived areas.PLOS ONE

Dear Dr. Lee,

Thank you for submitting your manuscript to PLOS ONE. After careful consideration, we feel that it has merit but does not fully meet PLOS ONE’s publication criteria as it currently stands. Therefore, we invite you to submit a revised version of the manuscript that addresses the points raised during the review process. Your revisions and responses have largely addressed the reviewer comments however one of the reviewers feels that there is scope from further clarification of the methods. This is an important area as it allows readers to judge the validity of your work, understand methodological limitations and, should they wish, to repeat the work themselves. 

We look forward to receiving your revised manuscript.

Kind regards,

Jenny Wilkinson, PhD

Academic Editor

PLOS ONE

Journal Requirements:

Reviewers' comments:

Reviewer's Responses to Questions

**Comments to the Author**

1. If the authors have adequately addressed your comments raised in a previous round of review and you feel that this manuscript is now acceptable for publication, you may indicate that here to bypass the “Comments to the Author” section, enter your conflict of interest statement in the “Confidential to Editor” section, and submit your "Accept" recommendation.

Reviewer #1: All comments have been addressed

Reviewer #2: (No Response)

2. Is the manuscript technically sound, and do the data support the conclusions?

Reviewer #1: Yes

Reviewer #2: Partly

3. Has the statistical analysis been performed appropriately and rigorously? 

Reviewer #1: Yes

Reviewer #2: N/A

4. Have the authors made all data underlying the findings in their manuscript fully available?

Reviewer #1: Yes

Reviewer #2: Yes

5. Is the manuscript presented in an intelligible fashion and written in standard English?

Reviewer #1: Yes

Reviewer #2: Yes

6. Review Comments to the Author

Reviewer #1: The authors have addressed my concerns around the inclusion of the inverse care law in both the introduction and the discussion

Reviewer #2: Many thanks for the updated review, I think restricting the inclusion criteria and making them more narrow has made this review clearer with the aims and results now much better aligning.

I still have a few concerns about the underlying methodology, some of this may just be making things clearer and explicit. From your review comments this whole search was repeated with new terms? And then a repeat synthesis done - that is quite a bit of work. You have said you changed the title screening to title and abstract screening on the flow chart - I presume you did this in real time too? Also I am still not clear on the screening process - were all articles screened separately by two reviewers, or a proportion. Am not sure what going through the title and abstract screening twice means - does that mean each article was reviewed twice or the same person did that section twice? You state both authors are happy with the included articles - I am not entirely sure what that means either? Did both of you do the search? All of this needs to be explicit, as it will allow readers to assess the quality of your work and strength of your findings.

You clearly have lost articles through changing your criteria, did your wider search terms elicit any new ones? I appreciate that you did not have access to an information scientist, may be helpful to state this as a limitation and what you did to mitigate it somewhere (sorry if you have and I missed that, I just couldn't see it in the article). Again no research can be methodologically perfect but it is important to be clear regarding what was and was not done.

Overall, the article reads much better and more clearly but the methodology still seems a bit unclear to me and I think the outstanding questions above need to be clarified.

7. PLOS authors have the option to publish the peer review history of their article (what does this mean?). If published, this will include your full peer review and any attached files.

Reviewer #1: **Yes: **Stewart Mercer

Reviewer #2: **Yes: **Marianne McCallum

---

## [Author Response · Author response to Decision Letter 1]

30 Oct 2024

Reviewer #1: The authors have addressed my concerns around the inclusion of the inverse care law in both the introduction and the discussion

Thank you for your assessment. Your feedback is appreciated in improving the quality of our work.

Reviewer #2: Many thanks for the updated review, I think restricting the inclusion criteria and making them more narrow has made this review clearer with the aims and results now much better aligning.

Thank you for your feedback. We appreciate that you find the overall clarity improved and we hope to have addressed your outstanding questions about the methodology in the revised manuscript below.

I still have a few concerns about the underlying methodology, some of this may just be making things clearer and explicit. From your review comments this whole search was repeated with new terms? And then a repeat synthesis done - that is quite a bit of work. You have said you changed the title screening to title and abstract screening on the flow chart - I presume you did this in real time too? 

Thank you for your feedback. To confirm, everything was done in real-time, and many hours (including weekends) were dedicated to this project. Both authors worked tirelessly on it, as we feel it is an important topic that deserves more attention, and are committed to getting it published for greater public awareness.

Also I am still not clear on the screening process - were all articles screened separately by two reviewers, or a proportion. Am not sure what going through the title and abstract screening twice means - does that mean each article was reviewed twice or the same person did that section twice? 

Thank you for your feedback. To clarify, the initial filtering process of the title and abstracts were performed twice by the same reviewer. The last filtering step of the full articles was performed by both reviewers and both reviewers were in agreement that the final thirteen articles included in the systematic review were of relevance to the topic. 

We have now changed the sentence under the section: ‘study selection’ to reflect this and make it clearer: 

“The first step of screening the article and abstracts were performed twice by one reviewer to ensure all the articles removed were on irrelevant topics or systematic reviews. Then, a filtering of the full text was performed where both reviewers collaborated and were in agreement that the selected articles included in the study were relevant to the topic of the systematic review.” 

You state both authors are happy with the included articles - I am not entirely sure what that means either? Did both of you do the search? All of this needs to be explicit, as it will allow readers to assess the quality of your work and strength of your findings.

Thank you for your feedback. One reviewer had performed the search and the initial filtering process of the tile and the abstracts twice. However, both reviewers collaborated on the final filtering of the full articles and agreed to include the final thirteen articles as they were deemed relevant to this systematic review. 

The sentence under the section ‘study selection’ was modified to communicate this and make it clearer: 

“The first step of screening the article and abstracts were performed twice by one reviewer to ensure all the articles removed were on irrelevant topics or systematic reviews. Then, a filtering of the full text was performed where both reviewers collaborated and were in agreement that the selected articles included in the study were relevant to the topic of the systematic review.” 

You clearly have lost articles through changing your criteria, did your wider search terms elicit any new ones? 

Thank you for your feedback. Yes, we can confirm the new search terms elicited five new articles which were included and discussed in the updated manuscript. 

I appreciate that you did not have access to an information scientist, may be helpful to state this as a limitation and what you did to mitigate it somewhere (sorry if you have and I missed that, I just couldn't see it in the article). Again no research can be methodologically perfect but it is important to be clear regarding what was and was not done.

Thank you for raising this. The lack of an information scientist has now been added to the limitations section of the systematic review: 

“Despite the vast majority of articles covered on PubMed and Embase, the decision to exclude other databases and the lack of access to an information scientist may have also slightly limited the breadth of this review.”

Overall, the article reads much better and more clearly but the methodology still seems a bit unclear to me and I think the outstanding questions above need to be clarified.

---

## [Editor Report · Decision Letter 2]

1 Nov 2024

PONE-D-24-36570R2A systematic review exploring the factors that contribute to increased primary care physician turnover in socio-economically deprived areas.PLOS ONE

Dear Dr. Lee,

Thank you for submitting your manuscript to PLOS ONE. After careful consideration, we feel that it has merit but does not fully meet PLOS ONE’s publication criteria as it currently stands. Therefore, we invite you to submit a revised version of the manuscript that addresses the points raised during the review process.

We look forward to receiving your revised manuscript.

Kind regards,

Jenny Wilkinson, PhD

Academic Editor

PLOS ONE

**Journal Requirements:**

**Additional Editor Comments:**

Thank you for your response and manuscript revisions. The reviewer comments have been addressed however there are a few minor presentation issues outstanding:

- for Table titles it is not necessary to state " Author created ...", it is sufficient to just give the title of the table

- in several places the reference number has been used in place of the author name (e.g. "(26) and (8) were classified ..." and "Whilst (23) utilised negative binomial regression, (24) employed ordinary least ..."), please check the manuscript and replace these citations with the author name(s) along with the reference number e.g. "Whilst Grigorglou et al. (23) utilised negative binomial regression, McGrail et al. (24) employed ordinary least ..." and citations given in square brackets rather than parenthesis

- please check the citation format in text to ensure that where there are several authors that et al is used e.g. in table 4 McGrail 2001 should be McGrail et al. 2001

- some of the citations in Table 4 and all those in Table 3 are missing a reference number

- under the Burnout Findings and Income Findings headings the text refers to 2 studies but 3 are listed in each case

---

## [Author Response · Author response to Decision Letter 2]

22 Nov 2024

Journal Requirements:

Thank you for raising this. We have reviewed the reference list, and we can confirm it is complete and correct. For the latest and final version, please refer to the revised manuscript without tracked changes (as EndNote keeps reverting to the old reference list when tracked changes are enabled).

Additional Editor Comments:

Thank you for your response and manuscript revisions. The reviewer comments have been addressed however there are a few minor presentation issues outstanding:

- for Table titles it is not necessary to state " Author created ...", it is sufficient to just give the title of the table

Thank you for your feedback. The title of every table has now been modified as per the instruction above: 

 “Table 1: Table detailing electronic databases searched, and number of papers identified.”

 “Table 2: Summary table of results”

 “Table 3: Summary table detailing how socioeconomic deprivation was measured for each article.”

 “Table 4: Visual depiction of ROBINS-I bias assessment”

- in several places the reference number has been used in place of the author name (e.g. "(26) and (8) were classified ..." and "Whilst (23) utilised negative binomial regression, (24) employed ordinary least ..."), please check the manuscript and replace these citations with the author name(s) along with the reference number e.g. "Whilst Grigorglou et al. (23) utilised negative binomial regression, McGrail et al. (24) employed ordinary least ..." and citations given in square brackets rather than parenthesis

Thank you for raising this. As requested, we have now replaced the reference number with the author name throughout the manuscript as seen in the following: 

“Furthermore, Dale et al. [9] found 82% of primary care physicians in England intended to: exit general practice, retire, take a career hiatus, or reduce their clinical hours within the upcoming five years.”

“Whilst Grigoroglou et al. [23] utilised negative binomial regression, McGrail et al. [24] employed ordinary least squares regression model, with Grigoroglou et al. [23] performing descriptive analyses to initially understand the basic distribution of the data.”

“The moderate risk of bias in McGrail et al. [14] stemmed from the absence of measurements related to professional factors, such as on-call arrangements, availability of locum relief, and infrastructure quality.”

“Parisi et al. [26] and Parisi et al. [8] were classified as having a moderate risk of bias due to the limitation of the covariates available within the national administrative datasets, and therefore were unable to cover all factors relevant to turnover.”

“Furthermore, McGrail et al. [24] compared Australia and the US, arguing that an increase in income did not encourage doctors to work in areas that had a shortage of primary care physicians.”

“Zhou et al. [30] did not find an association between socioeconomic deprivation and burnout, measured by emotional exhaustion and depersonalisation.”

“Ding et al. [22] emphasises this point and found that the position of salaried GPs which includes part time work and more flexible hours have reduced burnout.”

“Blane et al. [20] showed how in Scotland, despite more practices in the more economically deprived deciles, there were fewer primary care physicians available.

“This trend has been evident in Taylor et al. [28] from 1990-1994 England, quantifying primary care physician turnover and migration where the relatively deprived areas lost 23 primary care physicians compared with the relatively over-provided areas that gained 3 primary care physicians over the study period” 

“McGrail et al. [25] highlighted the increased mobility in remote USA areas, as they lacked professional support and hospital resources.”

“This is furthered by Parisi et al. [26], where the most deprived areas had a higher likelihood of GP turnover compared with practices located in the least deprived areas in England.”

“However, Taylor et al. [31] reviewed retention rates in 1991-2 England, and suggested an initial high retention rate of 85% for young new entrant GPs.”

“The Taylor et al. [31] 1991-92 study noted the large impact that a loss of a small number of GPs had on local health supply, with larger GP practices reducing the likelihood of younger GPs working.”

“Furthermore, unlike the findings of Vanasse et al. [29], this research indicated the proportion of ethnic minorities did not appear to affect retention rates. This discrepancy may be attributed by the context of the USA where Vanasse et al. [29] was conducted, as it identified a potential link between poverty and a higher proportion of ethnic minority populations, which could influence retention.”

“Chevillard & Mousques [15] highlights the role of PCTs in addressing geographical inequalities of GPs by reducing disparities in underserved areas.”

“Similarly, Chevillard et al. [21] comments within deprived rural regions, the implementation of PCTs have helped gain an average of 3.5 GPs per 100,000, improving healthcare accessibility.”

- please check the citation format in text to ensure that where there are several authors that et al is used e.g. in table 4 McGrail 2001 should be McGrail et al. 2001

Thank you for raising this. This change has been made throughout the manuscript (as seen above) and in all the relevant tables: 2, 3 and 4.

- some of the citations in Table 4 and all those in Table 3 are missing a reference number

Thank you for raising this. The references in table 3 and 4 have now been complete as seen below on the next page: 

Article Author

How was socioeconomic deprivation measured?

Blane et al. [19]

Scottish Index of Multiple Deprivation [SIMD]: practices were divided into deciles based on the percentage of patients in the top 15% of the most deprived postcodes weighted by population and overall SIMD score. 

Chevillard & Mosques [15]

Living areas classified into 6 sub-groups; 2 out of which were medically underserved. 

One group was characterised by a younger population with a higher amount of blue-collar workers and average access to healthcare. 

The second with rural and deprived areas with a greater proportion of elderly and blue-collar workers 

Chevillard et al. [20]

Cluster 1: suburban areas with a younger population underserved by GPs and nurses despite good socio-economic status [education, occupation and income]. 

Cluster 2: increased socio-economic status areas with an average level of accessibility of primary health care professionals. 

Cluster 3: industrial and agricultural areas with lower accessibility of primary health care professionals 

Cluster 4: low socio-economic status populations with lower accessibility of primary health care professionals.

Cluster 5: tourist areas with greatest accessibility of primary health care 

Cluster 6: remote areas with an older population. Accessibility of primary care is close to the average, but the GPs are older 

Ding et al. [21]

2004 Index of Multiple Deprivation 

Grigoroglou et al. [22]

Deprivation scores from the Index of Multiple Deprivation [IMD] were assigned to practices based on their postcode. The IMD is a measure of deprivation for the 32,844 small areas: Lower Layer Super Output Area [LSOAs] in England. Each area was given a score between 0 and 100, with higher scores indicating higher levels of deprivation.

McGrail et al. [23]

 Practices were classified according to the 9 level Rural Urban Continuum Codes and combined with 3 population groupings [<2500, 2500-20000, >20000] and adjacency to metropolitan area.

Deprivation was calculated using household income and median house prices for reference.

McGrail et al. [14] 

 Provider to population ratios calculated. 

Deprivation was also measured using US Medically Underserved Areas and Health Professional Shortage Areas and Australia’s District of Workforce Shortage Areas and Areas of Need 

Parisi et al. [24]

Index of Multiple Deprivation categorised in quintiles 

Parisi et al. [8]

Index of Multiple Deprivation and size of practice population was calculated in quintiles 

Taylor et al. [25]

Used deprivation bands 1-3. Also used Hacking’s measure of relative under/overprovision of GP services weighted for need 

Taylor et al. [26]

Deprivation was measured using the health authority deprivation score [UPA91] and the proportion of GP’s patients that were on deprivation payments 

Health authority level deprivation was measured by the Jarman index, also known as the Underprivileged Area score, which takes into account unemployment, overcrowded housing, single parent households, elderly, children under five, ethnic minorities, poverty and residential mobility.

Vanasse et al. [27]

Deprivation was measured via the mean per capita income of the population, [the ratio of hospital beds per 1000 population] and the rural or urban status of the geographic location 

Zhou et al. [28]

Deprivation was calculated using the Index of Multiple Deprivation quintiles which included the seven domains of: income, employment, health, living environment, barriers to housing, services and crime rates

Table 4: Visual depiction of ROBINS-I bias assessment 

Author/

Year 

 Bias due to confounding Bias in selection of participants into the study Bias due to deviations from intended interventions Bias due to missing data Bias in measurement of outcomes Bias in selection of the reported result Overall bias

Blane 2015

[19]

Chevillard 2021[15]

Chevillard 2019 [20]

Ding 2008 [21]

Grigoroglou 2022 [22]

McGrail 2017 [23]

McGrail 2017 [14]

Parisi 2021

[24]

Parisi 2023

[8]

Taylor 1998

[25]

Taylor 1999 [26]

Vanasse 2007 [27]

Zhou 2022 [28]

Notes: Assessment in accordance with ROBINS-I tool [18]: low risk of bias moderate risk of bias 

- under the Burnout Findings and Income Findings headings the text refers to 2 studies but 3 are listed in each case

Thank you for raising this. This has now been corrected to: 

Income findings: 

Two studies measured income influencing turnover and retention [23, 24] .

Burnout findings: 

Two studies [21, 28] agreed burnout was the main factor influencing retention and turnover of primary care physicians.

---

## [Editor Report · Decision Letter 3]

26 Nov 2024

A systematic review exploring the factors that contribute to increased primary care physician turnover in socio-economically deprived areas.

PONE-D-24-36570R3

Dear Dr. Lee,

We’re pleased to inform you that your manuscript has been judged scientifically suitable for publication and will be formally accepted for publication once it meets all outstanding technical requirements.

Kind regards,

Jenny Wilkinson, PhD

Academic Editor

PLOS ONE
---

## [Editor Report · Acceptance letter]

10 Dec 2024

PONE-D-24-36570R3 

PLOS ONE

Dear Dr. Lee, 

I'm pleased to inform you that your manuscript has been deemed suitable for publication in PLOS ONE. Congratulations! Your manuscript is now being handed over to our production team.

Kind regards, 

on behalf of

Dr Jenny Wilkinson 

Academic Editor

PLOS ONE